# Quantitative Estimation of Target Task Performance from Unsupervised Pretext Task in Semi/Self-Supervised Learning

**Lin-Han Jia** [1]   **Si-Yu Han** [1 2]   **Wen-Chao Hu** [1 2]   **Jie-Jing Shao** [1 2]
**Wen-Da Wei** [1 2]   **Zhi Zhou** [1 2]   **Lan-Zhe Guo** [1 3]   **Yu-Feng Li** [1 2]

## Abstract

The effectiveness of unlabeled data in Semi/Self-Supervised Learning (SSL) depends on appropriate assumptions for specific scenarios, thereby enabling the selection of beneficial unsupervised pretext tasks. However, existing research has paid limited attention to assumptions in SSL, resulting in practical situations where the compatibility between the unsupervised pretext tasks and the target scenarios can only be assessed after training and validation. This paper centers on the assumptions underlying unsupervised pretext tasks and explores the feasibility of preemptively estimating the impact of unsupervised pretext tasks at low cost. Through rigorous derivation, we show that the impact of unsupervised pretext tasks on target performance depends on three factors: assumption learnability with respect to the model, assumption reliability with respect to the data, and assumption completeness with respect to the target. Building on this theory, we propose a low-cost estimation method that can quantitatively estimate the actual target performance. We build a benchmark of over one hundred pretext tasks and demonstrate that estimated performance strongly correlates with the actual performance obtained through large-scale training and validation.

## 1. Introduction

In training machine learning models, a substantial amount of labeled data is typically required to achieve strong performance on target tasks. However, in practical scenarios, obtaining labeled data is often costly and labor-intensive. Consequently, effectively leveraging unlabeled data has become crucial for enhancing model performance under limited supervision.

Semi-supervised learning (Oliver et al., 2018; Van Engelen & Hoos, 2020) and self-supervised learning (Chen et al., 2020; Jing & Tian, 2020; Gui et al., 2024) (SSL) are two major paradigms for leveraging unlabeled data to enhance model performance on target tasks. The key difference lies in their training strategies: semi-supervised learning jointly trains on labeled and unlabeled data, whereas self-supervised learning first pre-trains on unlabeled data before fine-tuning on labeled data. Although they are often studied separately, both share a common theoretical foundation: training models on the pretext task using unlabeled data can influence their performance on the target task, and the design of such pretext tasks relies on prior assumptions (Rigollet, 2007; Mey & Loog, 2022; Liu et al., 2021; Lee et al., 2021; Misra & Maaten, 2020; Teng et al., 2022). For example, if the prior assumption is that the label of a sample remains invariant under rotation, a pretext task can be designed by enforcing the model to produce consistent predictions across rotated versions of the same sample. Essentially, both consistency regularization in semi-supervised learning and contrastive learning in self-supervised learning exploit such invariance to optimize target task performance.

The performance of SSL on target tasks heavily depends on the choice of pretext tasks, which essentially hinges on the underlying prior assumption. When the pretext task is poorly chosen, unlabeled data may yield no performance improvement or even lead to significant degradation (Oliver et al., 2018; Guo et al., 2020; Li et al., 2021; Jia et al., 2023; Lee et al., 2021; Kundu et al., 2020; Huang et al., 2023a). For example, the rotation invariance assumption cannot be applied to handwritten digit recognition datasets.

At present, research on how to estimate the benefit of a pretext task before training remains highly limited. Consequently, in practical applications, one must often conduct large-scale training and validation to assess whether a pretext task aligns with the target task. This process is not only delayed and computationally expensive but also requires

---

[1]National Key Laboratory for Novel Software Technology, Nanjing University [2]School of Artificial Intelligence, Nanjing University [3]School of Intelligence Science and Technology, Nanjing University. Correspondence to: Yu-Feng Li <liyf@lamda.nju.edu.cn>, Lan-Zhe Guo <guolz@lamda.nju.edu.cn>.

*Proceedings of the 43$^{rd}$ International Conference on Machine Learning*, Seoul, South Korea. PMLR 306, 2026. Copyright 2026 by the author(s).

the use of scarce labeled data for validation. If an effective method could be developed to estimate the expected test performance based on the used pretext task before training and validation, it would significantly reduce trial-and-error costs and enable the selection of the most beneficial unsupervised pretext task at minimal expense.

Since the performance of SSL essentially stems from the prior assumptions underlying its pretext tasks, the model injects these assumptions as knowledge through the use of unlabeled data (Lee et al., 2021; Huang et al., 2023a). We analyze this phenomenon through the lens of knowledge-integrated machine learning, particularly drawing on neuro-symbolic (Nesy) theories (Jia et al., 2025; He & Li, 2025; Hitzler & Sarker, 2022; Bhuyan et al., 2024; Shao et al., 2026; Yu et al., 2026). Unlike traditional knowledge-integrated settings, however, the prior assumptions in SSL are not entirely reliable—they hold only with certain probabilities. We therefore extend the theoretical framework to settings where knowledge is probabilistically reliable, thereby unifying the theories of knowledge-integrated learning and semi/self-supervised learning. Through rigorous theoretical analysis, we demonstrate that the benefit of training on unlabeled data with an assumption depends primarily on three factors: assumption learnability with respect to the model — the extent to which the model can satisfy the assumption through learning; assumption reliability with respect to the data — the extent to which the assumption holds true for the real data distribution; assumption completeness with respect to the target — the extent to which satisfying the assumption contributes to achieving the target task. In summary, from a theoretical standpoint, estimating the learnability, reliability, and completeness of a prior assumption is sufficient to estimate the model's generalization performance on the target task after training on the corresponding pretext task.

Building upon the theoretical analysis above, we further investigate how the three theoretical indicators can be transformed into empirically estimable quantities in practical applications. To this end, we propose a feasible approach that enables low-cost estimation of these three indicators. For learnability, we estimate the degree to which the prior assumption can be fitted by the model. Specifically, we train a model $\hat{f}_{\text{pretext}}$ on the pretext task using a small set of unlabeled data, and then evaluate the consistency between its predictions and the assumption after training. For reliability, we estimate the degree to which the prior assumption holds for the data. This requires access to the true labels of the unlabeled data; therefore, we approximate it by training a supervised model $\hat{f}_{\text{target}}$ on a small labeled subset as a surrogate oracle model to provide pseudo-labels for estimation. For completeness, we estimate the degree to which satisfying the prior assumption contributes to achieving the target task. This can be measured by comparing the pre-

diction consistency between $\hat{f}_{\text{pretext}}$ (trained to satisfy the assumption) and $\hat{f}_{\text{target}}$ (trained to achieve the target). Finally, by taking these three estimated indicators, this allows an estimation of target generation performance reflecting how beneficial a given pretext task is for the target task before large-scale training and label-based validation. The previously expensive process of selecting the most beneficial pretext task according to specific application scenarios becomes feasible.

We also constructed a benchmark consisting of over one hundred pretext tasks to evaluate the capability of the estimation algorithm to estimate target task performance before training and validation. Across these various tasks, we compared the estimated performance with the true performance obtained after full-scale semi/self-supervised training and validation. Experimental results show that, even when using only 5 labeled samples and 50 unlabeled samples per class for rapid estimation, the estimated performance is highly correlated with the actual performance. This demonstrates that the benefit of pretext tasks can be effectively assessed at a very low cost.

## 2. Related Work

### 2.1. Semi-Supervised Learning

Semi-supervised learning investigates how to jointly leverage unlabeled and labeled data to mitigate the issue of limited labels. Although consistency-based methods that rely heavily on strong data augmentations currently achieve state-of-the-art performance on public datasets (Berthelot et al., 2020; Xie et al., 2020; Sohn et al., 2020), they often lack sufficient generalization ability and may perform catastrophically in real-world scenarios (Oliver et al., 2018; Su et al., 2021; Jia et al., 2024). Researchers have increasingly uncovered that semi-supervised learning suffers from limited safety (Li & Zhou, 2014; Guo et al., 2020) and robustness (Huang et al., 2021; Mo et al., 2022; Jia et al., 2023). Examining the issue from first principles, the root of the safety and robustness problem lies in the inherent drawbacks of assumptions often fail to hold in the complex and variable conditions of real-world applications.

Although extensive research has been conducted on the robustness and safety of semi-supervised learning, it still fails to prevent cases in real-world applications where incorporating unlabeled data yields no performance improvement—or even leads to significant degradation (Oliver et al., 2018; Wang et al., 2022; Huang et al., 2023b; Jia et al., 2024). This is primarily because prior work has not systematically or scientifically analyzed the impact of unsupervised tasks on the target task, which is precisely the problem our work aims to address.

## 2.2. Self-Supervised Learning

Self-supervised learning has emerged as a powerful paradigm for learning useful representations from unlabeled data by constructing pretext tasks that generate supervision signals from the data itself. More recent advances have shifted toward contrastive learning frameworks, such as SimCLR (Chen et al., 2020), MoCo (He et al., 2020), and InfoNCE-based (Oord et al., 2018) methods, which aim to maximize agreement between different augmentations of the same sample while pushing apart representations of different samples. These methods demonstrated that large-scale self-supervised learning can approach or even surpass supervised learning on several downstream benchmarks. Beyond above, BYOL (Grill et al., 2020), SimSiam (Chen & He, 2021), Barlow Twins (Zbontar et al., 2021), and Dino (Caron et al., 2021) avoid negative samples altogether and still achieve competitive results, raising new questions about the underlying principles of representation learning.

Recent studies have identified a key challenge known as pretext task misalignment—where the objectives or representations learned during the pretext task do not align well with those required for the target task (Kundu et al., 2022; Bordes et al., 2024; you). In such cases, models trained with misaligned pretext tasks may even underperform compared to training from scratch or using only labeled data. This has motivated efforts to evaluate the transferability of SSL representations (Kornblith et al., 2019). In the theoretical research of self-supervised learning, there are also related works discussing the causal relationship between pretext tasks and target tasks (Lee et al., 2021; Huang et al., 2023a; Teng et al., 2022). However, despite these advances, estimating especially quantifying the effectiveness of a pretext task on a downstream target task remains an open challenge.

## 2.3. Knowledge-Integrated Machine Learning

Due to the problems of pure data-driven machine learning—namely its excessive dependence on the quantity, quality, and distribution of data—the new generation of machine learning research increasingly emphasizes paradigms jointly driven by knowledge and data (Von Rueden et al., 2021). In particular, neuro-symbolic learning (NeSy), which investigates how to incorporate prior knowledge into the learning process, has become a crucial area of study, yielding numerous theoretical and algorithmic advances (Yu et al., 2023; Hitzler & Sarker, 2022; Bhuyan et al., 2024). These advances include data-driven paradigms assisted by knowledge (Mao et al., 2019; Amayuelas et al., 2022; Morishita et al., 2023), knowledge-driven learning paradigms assisted by data (Getoor & Taskar, 2007; De Raedt & Kimmig, 2015; Wang et al., 2018), and paradigms in which knowledge and data interact and jointly drive learning (Xu et al., 2018; Manhaeve et al., 2018; Zhou, 2019; Petersen et al., 2021;

Stammer et al., 2023).

Most importantly, work in the NeSy field has examined the completeness of knowledge, highlighting that even when a model satisfies the given knowledge, it may still fail to produce correct predictions on the target task—a phenomenon known as the shortcut problem. The shortcut problem have highlighted when the knowledge bases are not sufficient, there will be multiple satisfying solutions, but the model fails to accurately pinpoint the target one. However, few works have provided viable solutions to address this incomplete problem.

In NeSy, knowledge is assumed to be entirely reliable, free from conflicts with the true data distribution. In contrast, in SSL, the prior assumptions are not reliable knowledge; they may not hold under the true data distribution, which can in turn lead to conflicts between the knowledge system and the data system.

# 3. Theory of Generalization Error Estimation Based on Unsupervised Pretext Tasks

## 3.1. Problem Formulation

In classical machine learning tasks, algorithms learn a model $f \in \mathcal{F}$ from a hypothesis space $\mathcal{F}$, using labeled data $D_L \sim \mathcal{D}_{x,y}$ drawn from the distribution $\mathcal{D}_{x,y}$ to achieve the target task $Task_{target}$. However, due to the limited availability of labeled data, this classical paradigm struggles to scale, leading to increasing interest in research areas such as semi-supervised learning and self-supervised learning, where pretext-tasks $Task_{pretext}$ are designed for unlabeled data $D_U \sim \mathcal{D}_x$ to assist in training. Nonetheless, theoretical studies on how training with $Task_{pretext}$ affects model performance on $Task_{target}$ are relatively scarce, resulting in a heavy reliance on heuristic experience for the selection of $Task_{pretext}$, and requiring a complete training and validation process to assess its effectiveness.

From the Bayesian perspective, machine learning is essentially the process of learning $p(y|x)$, whereas using unlabeled data only provides information about $p(x)$, which by itself does not contribute to learning $p(y|x)$. The reason why $Task_{pretext}$ can be helpful for $Task_{target}$ lies in its implicit introduction of prior knowledge based on assumption $\mathcal{K}_A$ during training. We denote by $((x, \mathcal{K}_A) \models \text{True})$ the compatibility between $x$ and the knowledge $\mathcal{K}_A$. Training on $Task_{pretext}$ enforces the model to satisfy $\mathcal{K}_A$, thereby forming a bridge to affect the target task via the relation:

$$p(y|x) = p[(x, \mathcal{K}_A) \models \text{True}|x] \qquad (1)$$
$$\cdot p[(y, \mathcal{K}_A) \models \text{True}|(x, \mathcal{K}_A) \models \text{True}, x]$$
$$\cdot p[y|(y, \mathcal{K}_A) \models \text{True}, (x, \mathcal{K}_A) \models \text{True}, x]$$

As a result, the model fits all of $\mathcal{K}_A$ during training of the

pretext task. Here, $p((x, \mathcal{K}_A) \models \text{True}|x)$ reflects the learnability of the knowledge by the model—that is, to what extent the model satisfies the knowledge; $p((y, \mathcal{K}_A) \models \text{True}|(x, \mathcal{K}_A) \models \text{True}, x)$ denotes the reliability of the knowledge with respect to the data—that is, to what extent the data conforms to the knowledge; $p(y|(y, \mathcal{K}_A) \models \text{True}, (x, \mathcal{K}_A) \models \text{True}, x)$ represents the completeness of the knowledge with respect to the target—that is, the extent to which the target depends on the knowledge.

In knowledge-integrated machine learning, the $\mathcal{K}_A$ is fully reliable, $p((y, \mathcal{K}_A) \models \text{True}|(x, \mathcal{K}_A) \models \text{True}, x) = 1$, which means that for a data point $(x, y)$, if $x$ satisfies the knowledge, then $y$ also satisfies the knowledge—there exists no data that violates the knowledge. However, in SSL, the $\mathcal{K}_A$ is probabilistically valid assumption which means $p((y, \mathcal{K}_A) \models \text{True}|(x, \mathcal{K}_A) \models \text{True}, x)$ is the probability that the assumption holds on the dataset.

Given a $Task_{pretext}$ and its corresponding assumed knowledge $\mathcal{K}_A$, if we can accurately estimate its learnability, reliability, and completeness, we can make an informed estimation about its performance on the target task.

### 3.2. Theoretical Derivation

Let $R_{target}$ denote the generalization error of the model on the target task:

$$R_{target}(f) = P_{x,y \sim \mathcal{D}_{x,y}}(f(x) \neq y) \tag{2}$$

Let $R_{unlearnable}$ denote the generalization error of the model on the unsupervised pretext task, which characterizes the learnability of the assumed prior assumption $\mathcal{K}_A$:

$$R_{unlearnable}(f, \mathcal{K}_A) = P_{x \sim \mathcal{D}_x}((f, x, \mathcal{K}_A) \not\models True) \tag{3}$$

Considering the case where the assumption is both reliable and complete, the pretext task is equivalent to the target task—i.e., satisfying the assumption is tantamount to accomplishing the target. Therefore, in the ideal case, the model's generalization error on the target task is equivalent to the learnability of the assumption associated with the pretext task. It thus follows that:

$$R_{target}(f) \leq R_{unlearnable}(f, \mathcal{K}_A) \tag{4}$$

Next, we consider the case where the assumption is not reliable. In this case, the generalization performance on the target task can no longer be determined solely by the model's degree of compliance with the assumed assumption $\mathcal{K}_A$. This is because data that do not actually satisfy the assumption may be incorrectly identified as satisfying it under the influence of $\mathcal{K}_A$, leading to an overestimation of the model's expected degree of assumption satisfaction relative to the true degree. The overestimated portion corresponds

to data that the learner fits as conforming to the assumption, while in fact they contradict it. To reflect the unreliability of the assumption over the data distribution, we define the following unreliability metric.

$$R_{unreliable}(f, \mathcal{K}_A) \tag{5}$$
$$= P_{x,y \sim \mathcal{D}_{x,y}, (f,x,\mathcal{K}_A) \models True}((y, \mathcal{K}_A) \not\models True)$$

This indicates that when the assumption is unreliable, the generalization error on the target task depends not only on the portion of the data where the assumption is unlearnable, but also on the portion where the assumption is learnable but inconsistent with the actual ground truth:

$$R_{target}(f) \leq R_{unlearnable}(f, \mathcal{K}_A) \tag{6}$$
$$+ (1 - R_{unlearnable}(f, \mathcal{K}_A)) \cdot R_{unreliable}(f, \mathcal{K}_A)$$

Then, we consider the case where the assumption is incomplete. In such cases, even if the model fully satisfies all true assumptions, it does not necessarily guarantee good performance on the target task. This indicates that multiple predictions may satisfy the given assumption, and more complete assumptions are required to distinguish the correct target predictions. This issue is a common form of the shortcut problem in Nesy. We define the incompleteness error as the portion where, despite the assumption being both learnable and consistent with the data, the model still fails to correctly predict the target.

$$R_{incomplete}(f, \mathcal{K}_A) \tag{7}$$
$$= P_{x,y \sim \mathcal{D}_{x,y}, (f,x,\mathcal{K}_A) \models True, (y,\mathcal{K}_A) \models True}(f(x) \neq y)$$

Finally, when the knowledge is potentially unreliable and incomplete, the generalization error on the target task is jointly determined by three factors: (1) the portion of the data where the knowledge is unlearnable, (2) the portion where the knowledge is learnable but inconsistent with the ground truth, and (3) the portion where the knowledge is both learnable and consistent with the data, yet still fails to yield accurate predictions. This leads to the following formulation:

$$R_{target}(f) \tag{8}$$
$$\leq R_{unlearnable}(f, \mathcal{K}_A) + (1 - R_{unlearnable}(f, \mathcal{K}_A))$$
$$\cdot R_{unreliable}(f, \mathcal{K}_A) + (1 - R_{unlearnable}(f, \mathcal{K}_A))$$
$$\cdot (1 - R_{unreliable}(f, \mathcal{K}_A)) \cdot R_{incomplete}(f, \mathcal{K}_A)$$
$$= 1 - (1 - R_{unlearnable}(f, \mathcal{K}_A)) \cdot (1 - R_{unreliable}(f, \mathcal{K}_A))$$
$$\cdot (1 - R_{incomplete}(f, \mathcal{K}_A))$$

## 4. Efficient Evaluation of the Impact of Pretext Tasks on Target Task Performance

Based on our theoretical results, we conclude that the generalization error of a model trained with an unsupervised

pretext task on a downstream target task is determined by the product of three theoretical factors: the learnability, reliability, and completeness of the underlying assumption. However, these theoretical indicators are typically difficult to compute directly in real-world applications. In our methods, we aim to approximate these three indicators using only a small amount of labeled data and unlabeled data, with minimal computational cost. This enables us to assess whether a given unsupervised pretext task is likely to benefit the current learning objective before training and validation. This section introduces the estimation methods for each of the three indicators.

### 4.1. The Estimation of Learnability

By definition, learnability describes the extent to which the assumption can be satisfied by the model. Since the objective of unsupervised pretext training is precisely to make the model conform to the underlying assumption, the degree of unsatisfiability essentially represents the model's general error on the pretext task which can be directly estimated by the training error on the pretext task, reflecting the model's ability to fit the prior assumption. Therefore, in this step, we directly train the model using a small sampled unlabeled dataset $\tilde{D}_U \sim D_U$ for efficiency considerations on the unsupervised pretext task whose loss function is denoted as $\text{Loss}_{\text{pretext}}$. Then we can obtain a model $\hat{f}_{\text{pretext}}$ that minimizes the pretext task error:

$$\hat{f}_{\text{pretext}} \tag{9}$$
$$= \arg\min_f \frac{1}{|\tilde{D}_U|} \sum_{x \in \tilde{D}_U} Loss_{\text{pretext}}(f, x, \mathcal{K}_A)$$

Subsequently, the training error of $\hat{f}_{\text{pretext}}$ on the sampled unlabeled dataset $\tilde{D}_U$ can be used as an estimate of $R_{\text{unlearnable}}$:

$$\hat{R}_{\text{unlearnable}} \tag{10}$$
$$= \frac{1}{|\tilde{D}_U|} \sum_{x \in \tilde{D}_U} \mathbb{I}((\hat{f}_{\text{pretext}}, x, \mathcal{K}_A) \not\models True)$$

For example, in consistency regularization for semi-supervised learning and contrastive learning for self-supervised learning, the pretext task involves applying two transformations to the model and minimizing the difference between their predictions. The corresponding prior assumption is that the label should remain unchanged after two different transformations of the same data. $\text{Loss}_{\text{pretext}}$ refers to the consistency loss or contrastive loss, depending on the SSL algorithm in use. $\hat{R}_{\text{unlearnable}}$ represents the proportion of unlabeled data for which the model still fails to achieve prediction consistency after consistency-based optimization. After this estimation, we proceed to the next metric by selecting the subset of unlabeled data for which

the knowledge is satisfied.

$$\tilde{D}_U^{learnable} \tag{11}$$
$$= \{x \in \tilde{D}_U \mid (\hat{f}_{\text{pretext}}, x, \mathcal{K}_A) \models True\}$$

### 4.2. The Estimation of Reliability

Reliability describes the extent to which the assumption is consistent with the data, with a particular focus on whether the assumption holds true on the current dataset. In semi-supervised or self-supervised learning, since the assumption originates from assumed priors, it is inherently valid only for a subset of the data. However, during unsupervised training, it is typically assumed that all data conform to the assumption. This mismatch may cause certain samples, whose true labels conflict with the assumption to be forcibly fitted by the model as if they satisfy the assumption, even though they do not. Such samples inevitably fail to contribute positively to model performance. To empirically estimate the degree to which the assumption is violated, we need to determine whether a conflict exists between the data's true label and the imposed assumption, denoted as:

$$\hat{R}_{\text{unreliable}} \tag{12}$$
$$= \frac{1}{|\tilde{D}_U^{learnable}|} \sum_{x \sim \tilde{D}_U^{learnable}} \mathbb{I}((y_x, \mathcal{K}_A) \not\models True)$$

Here, $y_x$ denotes the true label of an unlabeled data sample $x$. However, since $y_x$ is unknown in practice, this poses a challenge for estimating $\hat{R}_{\text{unreliable}}$. Ideally, we would expect to have access to an oracle model that can directly provide the true label $y_x$ for any unlabeled sample $x$. Given the impracticality of such an oracle, we approximate it by training a model $\hat{f}_{\text{target}}$ on a small sampled labeled dataset $\tilde{D}_L \sim D_L$ corresponding to the target task whose loss function is denoted as $\text{Loss}_{\text{target}}$. In reality, when the scale of $D_L$ is relatively small, $\tilde{D}_L = D_L$ can be directly adopted. The model $\hat{f}_{\text{target}}$ serves as a practical substitute for the oracle model during evaluation, i.e.,

$$\hat{f}_{\text{target}} = \arg\min_f \frac{1}{|\tilde{D}_L|} \sum_{x,y \in \tilde{D}_L} Loss_{\text{target}}(f(x), y) \tag{13}$$

Consequently, this enables a more refined empirical estimation of $\hat{R}_{\text{unreliable}}$, given by:

$$\hat{R}_{\text{unreliable}} \tag{14}$$
$$= \frac{1}{|\tilde{D}_U^{learnable}|} \sum_{x \sim \tilde{D}_U^{learnable}} \mathbb{I}((\hat{f}_{\text{target}}(x), \mathcal{K}_A) \not\models True)$$

After completing the estimation, we select the data samples whose predicted true labels are consistent with the

knowledge base, and use them for the estimation of the next indicator, i.e.,

$$\tilde{D}_U^{reliable} \tag{15}$$
$$= \{ x \in \tilde{D}_U^{learnable} \mid (\hat{f}_{target}(x), \mathcal{K}_A) \models True \}$$

### 4.3. The Estimation of Completeness

By definition, Completeness describes the extent to which the assumption is sufficient for the target task, focusing on whether the assumption adequately covers the target. In the ideal case, the pretext task is identical to the target task, or the target task is a subset of the pretext task. In such scenarios, successfully solving the pretext task is equivalent to accomplishing the target, and the prior assumption associated with the pretext task is considered complete. However, in practice, even if the model fully learns the given assumption, it may still fall short of achieving the target task perfectly. Completeness quantifies the extent to which the target can be achieved using the reliable assumption. To empirically estimate completeness, we need to assess whether the model trained on the pretext task truly has the potential to solve the target task. As before, since the true label $y_x$ is unknown, we approximate it using $\hat{f}_{\text{target}}(x)$. More intuitively, training $\hat{f}_{\text{pretext}}$ aims to satisfy the assumption, while training $\hat{f}_{\text{target}}$ aims to achieve the target, So the completeness of the assumption with respect to the target can be approximated by the consistency between $\hat{f}_{\text{pretext}}$ and $\hat{f}_{\text{target}}$,

$$\hat{R}_{incomplete} \tag{16}$$
$$= \frac{1}{|\tilde{D}_U^{reliable}|} \sum_{x \sim \tilde{D}_U^{reliable}} \mathbb{I}(\hat{f}_{target}(x) \neq \hat{f}_{pretext}(x))$$

It is worth noting that $\hat{f}_{\text{pretext}}$ is the model trained solely on the pretext task, and its prediction targets are not inherently aligned with those of the actual target task. Therefore, for each class, we use $\hat{f}_{\text{pretext}}$ to extract the embeddings of samples in $\tilde{D}_L$ belonging to that class and compute the centroid of all these embeddings as the prototype for the class. For unlabeled data $x \sim \tilde{D}_U$, we use $\hat{f}_{\text{pretext}}$ to extract its embedding, and the predicted label for $\hat{f}_{\text{pretext}}(x)$ is the one whose prototype has the closest Euclidean distance to this embedding. This process helps to align $\hat{f}_{\text{pretext}}$ with $\hat{f}_{\text{target}}$ in terms of the target task.

The model's test performance after full semi-supervised or self-supervised learning can ultimately be estimated based on the three estimated indicators:

$$\hat{R}_{target} = 1 - (1 - \hat{R}_{unlearnable}) \tag{17}$$
$$\cdot (1 - \hat{R}_{unreliable}) \cdot (1 - \hat{R}_{incomplete})$$

## 5. Experiments

To verify whether the estimated target performance based on empirical indicators of assumption learnability, reliability, and completeness can reflect the model's actual performance after large-scale semi-supervised or self-supervised training, we constructed a benchmark consisting of over one hundred pretext tasks. Under different pretext tasks, we compared the quickly estimated performance based on theoretical results with limited data against the actual performance of models trained with a large amount of data using full semi-supervised or self-supervised procedures. To minimize estimation cost, we sampled only a very small subset from the large pool of unlabeled data for prediction in low-label scenarios. The experimental results align with our expectations, demonstrating that, regardless of whether the setting is semi-supervised learning (where unsupervised pretext tasks and supervised target tasks are trained jointly) or self-supervised learning (where unsupervised pretext task pretraining is followed by fine-tuning on the supervised target task), the model's actual performance is highly correlated with our theoretical estimates based on limited data.

### 5.1. Basic Experimental Setting

We selected 11 commonly used unsupervised pretext tasks in current self-supervised and semi-supervised learning settings, and applied different augmentation strength parameters, ultimately constructing a benchmark containing 115 unsupervised pretext tasks which are shown in Table 1. All operations in the table are implemented based on the torchvision.transforms module. Under the same dataset, model, and hyperparameters, we used different pretext tasks to perform both performance estimation and actual training. These unsupervised pretext tasks include ResizedCrop, Rotation, Translate, Shear, Scale, Brightness, Contrast, Saturation, Hue, HorizontalFlip, and VerticalFlip—all of which are widely adopted in consistency-based semi-supervised learning (Tarvainen & Valpola, 2017; Xie et al., 2020; Sohn et al., 2020) and contrastive self-supervised pretraining (Chen et al., 2020; He et al., 2020; Grill et al., 2020; Caron et al., 2021). The prior assumption introduced by these pretext tasks assumes that the label remains consistent after data transformations; specifically, the model is expected to produce identical predictions for two independently augmented views of the same input sample.

Under the basic setup, we conducted experiments on the CIFAR-10, CIFAR-100 (Krizhevsky et al., 2009) and ImageNet-200 (Yang, 2015) datasets. For each class, we selected only 5 labeled examples—i.e., only 50 labeled samples in total for CIFAR-10, 500 for CIFAR-100 and 1000 for ImageNet-200. For performance estimation based on proposed indicators, we used 50 unlabeled examples per class—i.e., 500 unlabeled samples for CIFAR-10, 5,000

*Table 1.* Candidate Pretext Tasks for Experiments

| Name | Strength | Operation |
|------|----------|-----------|
| ResizedCrop | $s \in [0,10] \cap \mathbb{Z}$ | ResizedCrop(scale=s/10) |
| Rotation | $s \in [0,10] \cap \mathbb{Z}$ | Rotation(degrees=s/10*180) |
| Translate | $s \in [0,10] \cap \mathbb{Z}$ | Affine(translate=s/10) |
| Shear | $s \in [0,10] \cap \mathbb{Z}$ | Affine(shear=s/10) |
| Scale | $s \in [1,10] \cap \mathbb{Z}$ | Affine(scale=s/10) |
| Brightness | $s \in [0,10] \cap \mathbb{Z}$ | ColorJitter(brightness=s/10) |
| Contrast | $s \in [0,10] \cap \mathbb{Z}$ | ColorJitter(contrast=s/10) |
| Saturation | $s \in [0,10] \cap \mathbb{Z}$ | ColorJitter(saturation=s/10) |
| Hue | $s \in [0,5] \cap \mathbb{Z}$ | ColorJitter(hue=s/10) |
| HorizonFlip | $s \in [0,10] \cap \mathbb{Z}$ | HorizontalFlip(p=s/10) |
| VerticalFlip | $s \in [0,10] \cap \mathbb{Z}$ | VerticalFlip(p=s/10) |

for CIFAR-100 and 10000 for ImageNet-200. In the actual semi-supervised and self-supervised training, all the training samples excepted labeled ones were used as unlabeled data—i.e., 49,950 samples for CIFAR-10 and 49,500 for CIFAR-100 and 99,000 for ImageNet-200. All evaluations of actual performance were conducted on the full test sets. Ultimately, we compared the estimated performance against the actual performance of self-supervised and semi-supervised learning.

In all experiments, we used ViT-B (Dosovitskiy, 2020) with a patch size of 16 as the baseline model. The batch size was set to 64 and the epoch is set to 5, the optimizer was Adam (Kingma, 2014), the learning rate was fixed at 5e-5, and the weight decay was set to 0.01. The loss function for unsupervised pretext tasks was uniformly set to mean squared error (MSE) loss, and the loss function for the supervised target task was set to cross-entropy loss. All experiments were conducted using the PyTorch framework on four A800 GPUs.

We use the Pearson correlation coefficient(Pearson, 1895) as the metric to evaluate the degree of correlation between estimated performance and actual performance. Under the same conditions, this metric can be used to assess the effectiveness of the pretext task evaluation method.

### 5.2. Experimental Results

Finally, we estimated the empirical indicators of assumption learnability, reliability, and completeness for all 115 unsupervised pretext tasks under limited data. Based on these three indicators, we computed the estimated target performance. We also obtained the actual performance through full semi-supervised and self-supervised learning processes evaluated on real test data. To intuitively illustrate the correlation between the estimated and actual performance, we plotted scatter diagrams. We represent operation using color and strength using opacity. All values on the axes are expressed as percentages. Figures 1 and 2 show,for

the CIFAR-10 dataset, the comparison between estimated performance and actual performance under semi-supervised and self-supervised settings, respectively. In terms of the final results, the Pearson correlation coefficients between estimated and actual performance for semi-supervised learning and self-supervised learning reach 0.785 and 0.820 respectively on CIFAR-10. The experimental results on the CIFAR-100 and ImageNet-200 are provided in the appendix.

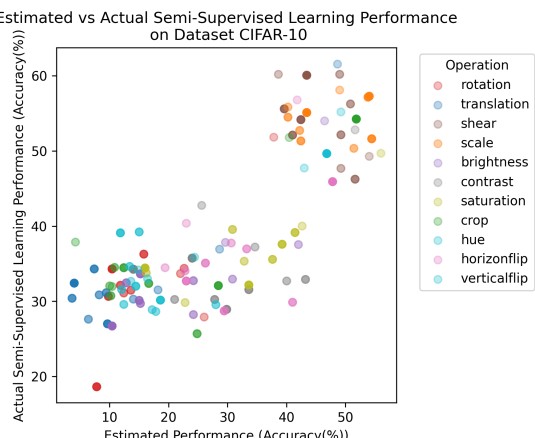

*Figure 1.* This figure illustrates the comparison between estimated performance and final actual performance after full semi-supervised learning on CIFAR10 with the basic setup.

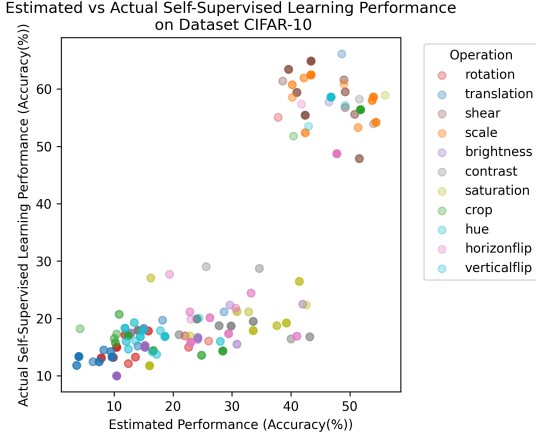

*Figure 2.* This figure illustrates the comparison between estimated performance and final actual performance after full self-supervised learning on CIFAR-10 with the basic setup.

From the experimental results, it is evident that whether using semi-supervised learning or self-supervised learning, the final test performance is highly correlated with the estimated performance derived from the three empirical theoretical indicators. As long as the same pretext task is used, When the pretext task changes, the performance of semi-supervised learning and self-supervised learning varies in an almost synchronized manner. This indicates that whether a pretext task can benefit the target task is not strongly influenced by the training strategy. Another noteworthy observation

is that the model's performance on the target task tends to be highly polarized. This reflects the cumulative effect of pretext tasks on neural networks: when a pretext task is beneficial to the target, its positive impact becomes amplified as training progresses; conversely, when a pretext task is detrimental, its negative impact is also amplified with more training, and intermediate cases rarely occur.

## 5.3. Ablation Study

To demonstrate that the target performance estimated jointly by the three proposed metrics is most correlated with the true target performance, we conducted ablation experiments. We verified the correlation between estimated and actual performance using the three indicators individually or in pairs respectively. The experimental results on CIFAR-10 under both semi-supervised and self-supervised settings are shown in Tables 2 and 3. The results of the ablation experiments on the CIFAR-100 and ImageNet-200 datasets are presented in the appendix.

*Table 2.* Ablation study on estimating self-supervised learning performance using the three indicators on the CIFAR-10 dataset.

| Learnability | Reliability | Completeness | Correlation |
|:---:|:---:|:---:|:---:|
| ✓ | ✗ | ✗ | 0.224 |
| ✗ | ✓ | ✗ | 0.704 |
| ✗ | ✗ | ✓ | 0.727 |
| ✓ | ✓ | ✗ | 0.716 |
| ✓ | ✗ | ✓ | 0.746 |
| ✗ | ✓ | ✓ | 0.808 |
| ✓ | ✓ | ✓ | **0.820** |

*Table 3.* Ablation study on estimating semi-supervised learning performance using the three indicators on the CIFAR-10 dataset.

| Learnability | Reliability | Completeness | Correlation |
|:---:|:---:|:---:|:---:|
| ✓ | ✗ | ✗ | 0.192 |
| ✗ | ✓ | ✗ | 0.686 |
| ✗ | ✗ | ✓ | 0.692 |
| ✓ | ✓ | ✗ | 0.694 |
| ✓ | ✗ | ✓ | 0.710 |
| ✗ | ✓ | ✓ | 0.773 |
| ✓ | ✓ | ✓ | **0.785** |

The results show that each metric is contributive: in both semi-supervised and self-supervised settings, adding any additional metric brings the estimated performance closer to the true performance, enabling a more accurate assessment of the impact of each unsupervised pretext task.

## 5.4. Examination of Robustness

To demonstrate that the method proposed in this paper can effectively estimate true performance across different experimental settings, we conducted multiple robustness tests:

- Algorithm robustness: Unlike the basic setup, where both semi-supervised and self-supervised learning use the simplest unified training algorithm—i.e., MSE loss based on consistency as $Loss_{\text{pretext}}$ and cross-entropy loss as $Loss_{\text{target}}$, we further validated the effectiveness of our method on algorithms specific to each domain. This includes MeanTeacher (Tarvainen & Valpola, 2017), UDA (Xie et al., 2020), and FixMatch (Sohn et al., 2020) in the semi-supervised domain, and Sim-CLR (Chen et al., 2020), MoCo (He et al., 2020), BYOL (Grill et al., 2020), and Dino (Caron et al., 2021) in the self-supervised domain. All semi-supervised learning algorithms follow the implementation of LAMDA-SSL, and all self-supervised learning algorithms follow the implementation of Lightly.

- Model robustness: Unlike the basic setup where ViT-B is used as the backbone, we further verified our method using the ResNet-50 (He et al., 2016) model.

- Data distribution robustness: Unlike the basic setup where labeled and unlabeled data share the same distribution, we further tested the method on the out-of-distribution dataset Office-31 (Saenko et al., 2010).

- Data scale robustness: Unlike the basic setup, where the unlabeled data used for performance estimation is 10 times the size of the labeled data, we further tested our method with unlabeled data amounts of 50 times the size of the labeled data.

- Hyperparameter robustness: Unlike the basic setup where the number of epochs is 5 and the learning rate is $5 \times 10^{-5}$, we further validated the method under settings with various settings.

All the above robustness tests indicate that the three-metric-based estimation method can effectively estimate true performance under diverse conditions. Detailed results can be found in the appendix.

## 6. Estimation Error Analysis

In the previous section, we established the theoretical relationship between the proposed metrics and the target downstream performance. We now further analyze the estimation error introduced by finite-sample approximation. Specifically, we study the gap between the empirical quantities $\hat{R}_{\text{unlearnable}}$, $\hat{R}_{\text{unreliable}}$, $\hat{R}_{\text{incomplete}}$ and their corresponding population-level quantities $R_{\text{unlearnable}}$, $R_{\text{unreliable}}$, $R_{\text{incomplete}}$.

To characterize the complexity of the hypothesis spaces, we introduce the Rademacher complexity framework. Let $\mathcal{F}_{\text{pretext}}$ and $\mathcal{F}_{\text{target}}$ denote the hypothesis spaces of the pretext model $\hat{f}_{\text{pretext}}$ and the target model $\hat{f}_{\text{target}}$, respectively. Their corresponding Rademacher complexities are denoted by $\mathfrak{R}(\mathcal{F}_{\text{pretext}}, n_U)$ and $\mathfrak{R}(\mathcal{F}_{\text{target}}, n_L)$, where $n_U = |\tilde{D}_U|$ and $n_L = |\tilde{D}_L|$ represent the sizes of the unlabeled and labeled sampled datasets.

Intuitively, the estimation error of $\hat{R}_{\text{unlearnable}}$ is governed by the complexity of the pretext model space and the amount of unlabeled data. Similarly, the estimation error of $\hat{R}_{\text{unreliable}}$ depends on the complexity of the target model space and the amount of labeled data. Since $\hat{R}_{\text{incomplete}}$ is jointly influenced by both pretext and target predictions, its estimation error accumulates the errors from both components.

**Theorem 6.1** (Estimation Error Bound of Learnability). *For any $\delta \in (0, 1)$, with probability at least $1 - \delta$, the estimation bias satisfies*

$$|\hat{R}_{unlearnable} - R_{unlearnable}| \leq \epsilon_{pretext}, \quad (18)$$

*where*

$$\epsilon_{pretext} = 2\mathfrak{R}(\mathcal{F}_{pretext}, n_U) + \sqrt{\frac{\log(2/\delta)}{2n_U}}. \quad (19)$$

The above theorem shows that the estimation accuracy of learnability improves as the unlabeled sample size increases or the complexity of the pretext hypothesis space decreases.

**Theorem 6.2** (Estimation Error Bound of Reliability). *For any $\delta \in (0, 1)$, with probability at least $1 - \delta$, the estimation bias satisfies*

$$|\hat{R}_{unreliable} - R_{unreliable}| \leq \epsilon_{target}, \quad (20)$$

*where*

$$\epsilon_{target} = 2\mathfrak{R}(\mathcal{F}_{target}, n_L) + \sqrt{\frac{\log(2/\delta)}{2n_L}}. \quad (21)$$

The above result indicates that the reliability estimation becomes tighter when the target hypothesis space is properly controlled or when more labeled data are available.

**Theorem 6.3** (Estimation Error Bound of Completeness). *For any $\delta \in (0, 1)$, with probability at least $1 - \delta$, the estimation bias satisfies*

$$|\hat{R}_{incomplete} - R_{incomplete}| \leq \epsilon_{pretext} + \epsilon_{target}. \quad (22)$$

Theorem 3 reveals that the completeness estimation error accumulates the uncertainty introduced by both the pretext and target models.

Finally, we analyze the propagated estimation error on the overall target risk.

**Theorem 6.4** (Estimation Error Bound of Target Risk). *For any $\delta \in (0, 1)$, with probability at least $1 - \delta$, the estimation bias satisfies*

$$|\hat{R}_{target} - R_{target}| \leq 2(\epsilon_{pretext} + \epsilon_{target}) + (\epsilon_{pretext} + \epsilon_{target})^2$$
$$+ \epsilon_{pretext} \cdot \epsilon_{target} \cdot (1 + \epsilon_{pretext} + \epsilon_{target}). \quad (23)$$

Overall, the above theoretical results establish a unified estimation error framework for the proposed metrics. The bounds demonstrate that the empirical estimators converge to their population counterparts as the sample size increases and the hypothesis complexity is properly controlled. Moreover, the final target estimation error is explicitly characterized by the combined uncertainties of the pretext and target learning processes, providing theoretical guarantees for the reliability of the proposed evaluation framework.

## 7. Hypothesis Testing

We performed t-tests on all experimental conclusions at a 99% confidence level to verify whether the estimated performance achieved at low cost by our proposed scheme is significantly positively correlated with the actual performance obtained through extensive consumption. Since we conducted experiments on a total of 115 pretext tasks, based on the t-distribution with degrees of freedom (df) = 115 - 2 = 113 and a one-tailed significance level ($\alpha$) = 0.01 corresponding to a significant positive correlation at the 99% confidence level, the critical Pearson correlation coefficient was calculated to be 0.226. In all experimental results of this paper, the correlation coefficients between the estimated performance and actual performance are far higher than this critical value, which sufficiently demonstrates that the proposed estimation method can accurately predict the real performance that relies on large-scale training and validation at an extremely low cost.

## 8. Conclusion

We conducted an in-depth investigation into how training on unsupervised pretext tasks incorporates prior knowledge or assumption into models. We demonstrate that, in theory, the impact of pretext tasks on target performance hinges on three factors: assumption learnability, reliability, and completeness. This reveals the connection between prior knowledge or assumption and the three fundamental components of machine learning—model, data, and target.

It is not difficult to foresee that in the future, the scale of models, data and targets will continue to grow, while reliable labels will not increase accordingly. This means that unsupervised training will occupy an increasingly significant role. As the starting point of training, the selection of pretext tasks based on prior knowledge will become increasingly important throughout the machine learning pipeline.

## Impact Statement

This paper presents work whose goal is to advance the field of Machine Learning. There are many potential societal consequences of our work, none which we feel must be specifically highlighted here.

## Acknowledgements

This research was supported by the Jiangsu Science Foundation (BK20243012,BG2024036,BK20232003), Natural Science Foundation of China (62576162), the Fundamental Research Funds for the Central Universities (022114380023), and China Computer Federation Doctoral Student Funding Program.

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

# A. Experiments on CIFAR-100

We conducted experiments on the CIFAR-100 dataset following the settings described in **??**, verifying the difference between estimated performance and actual performance. We also plotted scatter plots and calculated the correlation coefficients between estimated performance and actual performance under semi-supervised and self-supervised settings, which are 0.905 and 0.950 respectively. The comparison results can be seen in Figures 3 and 4.

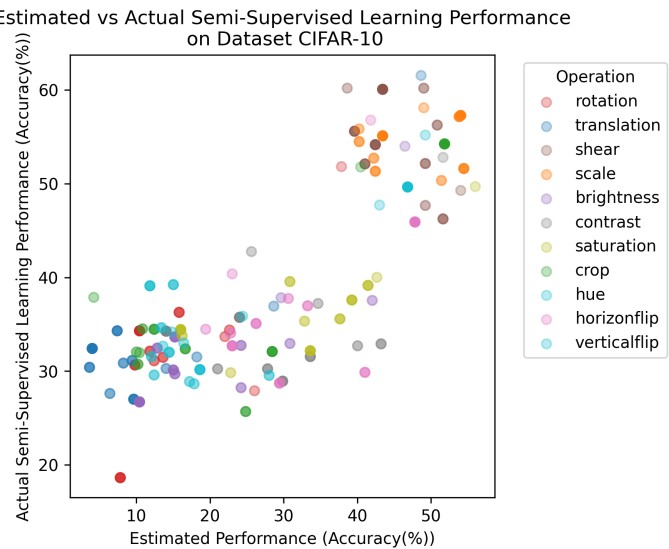

*Figure 3.* This figure illustrates the comparison between estimated performance and final actual performance after full semi-supervised learning on CIFAR-100 with the basic setup.

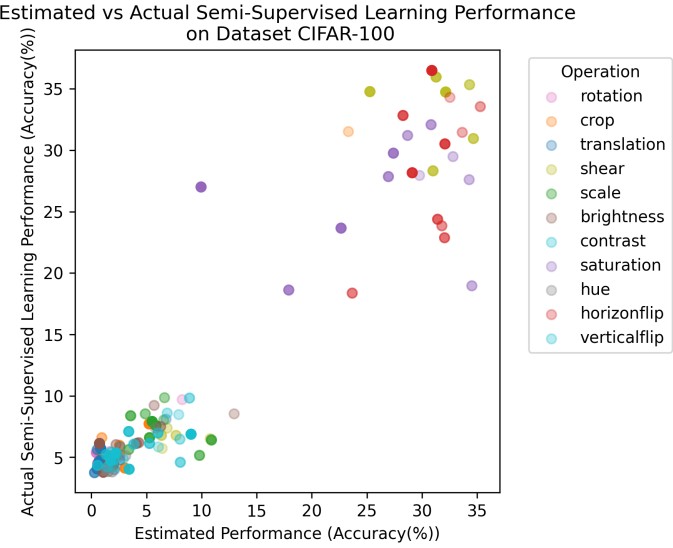

*Figure 4.* This figure illustrates the comparison between estimated performance and final actual performance after full self-supervised learning on CIFAR-100 with the basic setup.

# B. Experiments on ImageNet-200

We conducted experiments on the ImageNet-200 dataset following the settings described in **??**, verifying the difference between estimated performance and actual performance. We also plotted scatter plots and calculated the correlation

coefficients between estimated performance and actual performance under semi-supervised and self-supervised settings, which are 0.675 and 0.678 respectively. The comparison results can be seen in Figures 5 and 6.

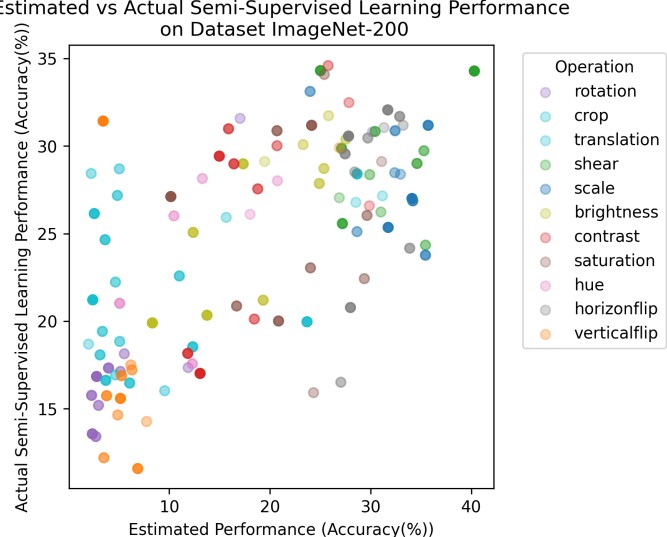

*Figure 5.* This figure illustrates the comparison between estimated performance and final actual performance after full semi-supervised learning on ImageNet-200 with the basic setup.

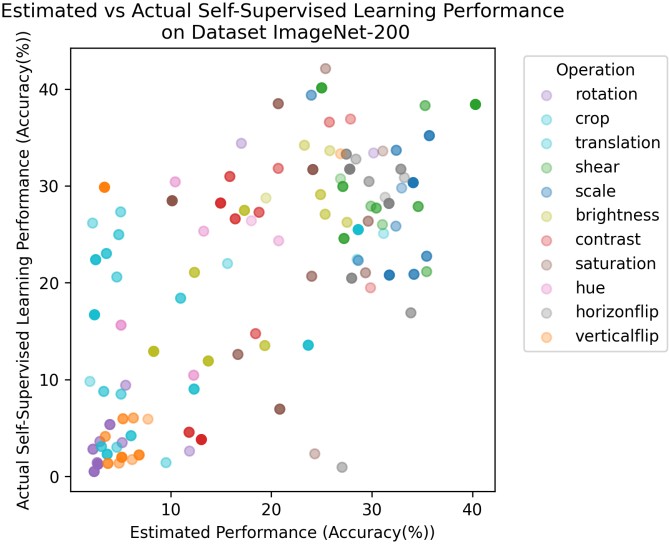

*Figure 6.* This figure illustrates the comparison between estimated performance and final actual performance after full self-supervised learning on ImageNet-200 with the basic setup.

## C. Experiments on the Algorithm Robustness

In the experiments under the basic settings, we unified the two paradigms of semi-supervised learning and self-supervised learning, conducted comparative experiments using a consistent loss function, and only distinguished them by the training order of the pretext task and the target task (for semi-supervised learning, the two tasks are trained simultaneously; for self-supervised learning, the pretext task is trained prior to the target task). However, we did not consider the differences between the two in existing classic algorithms. To address this, we adopted representative classic algorithms from each

of the two fields and compared the predicted performance with the actual performance obtained through training on these classic algorithms.

For semi-supervised learning, we compared against the actual performance trained on three classic algorithms: MeanTeacher, UDA, and FixMatch. Except for the algorithms themselves, the backbone model, hyperparameters, dataset settings, and other configurations remained consistent with the basic settings. We also plotted scatter plots and calculated the correlation coefficients between estimated performance and actual performance under 3 algorithms, which are 0.801, 0.747, and 0.902 respectively. The comparison results can be seen in Figures 7 to 9.

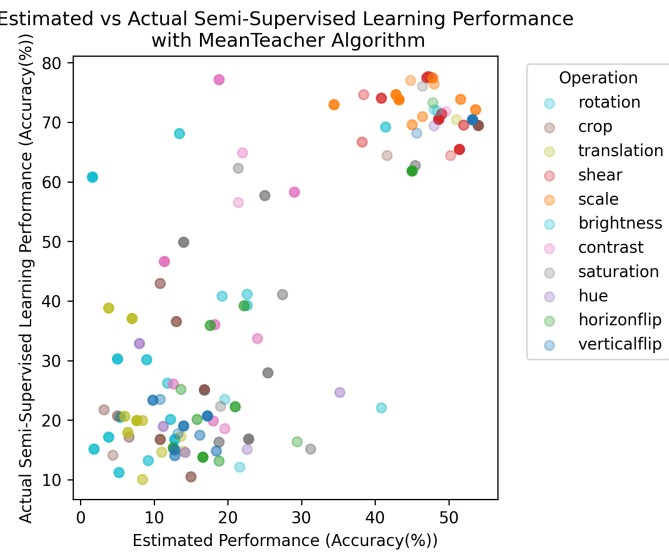

*Figure 7.* This figure illustrates the comparison between estimated performance and final actual performance after full semi-supervised learning on CIFAR-10 with the MeanTeacher Algorithm.

For self-supervised learning, we compared against the actual performance trained on four classic algorithms: SimCLR, MOCO, BYOL, and DINO. Except for the algorithms themselves, the backbone model, hyperparameters, dataset settings, and other configurations remained consistent with the basic settings. We also plotted scatter plots and calculated the correlation coefficients between estimated performance and actual performance under 4 algorithms, which are 0.557, 0.453, 0.603 and 0.668 respectively. The comparison results can be seen in Figures 10 to 13.

## D. Experiments on the Model Robustness

In the basic settings, we adopted the Vision Transformer (ViT) model as the backbone. As a representative of current transformer-based models, ViT boasts strong generality. To verify that the proposed estimation method is also applicable to other backbones, we conducted identical experiments on ResNet-50 models. We also plotted scatter plots and calculated the correlation coefficients between estimated performance and actual performance under semi-supervised and self-supervised settings, which are 0.478 and 0.694 respectively. The comparison results can be seen in Figures 14 and 15.

The comparison results can be seen in Figures 14 and 15.

## E. Experiments on the Data Distribution Robustness

Since self-supervised learning often utilizes unlabeled data with a different distribution from labeled data for pre-training, we further validated our results on the cross-distribution dataset Office-31—unlike the same-distribution datasets (CIFAR-10, CIFAR-100, and ImageNet-200) used in the basic settings. The results demonstrate that our proposed estimation method remains effective when the labeled data and unlabeled data are from different distributions in the pre-training and fine-tuning paradigm. We use data with the domain of Amazon as labeled data, and data with the domains of Dslr and Webcam as unlabeled data. 155 labeled data points (5 per category) and 1,550 unlabeled data points (50 per category) are used for performance estimation. We also plotted the scatter plot and calculated the correlation coefficient between estimated

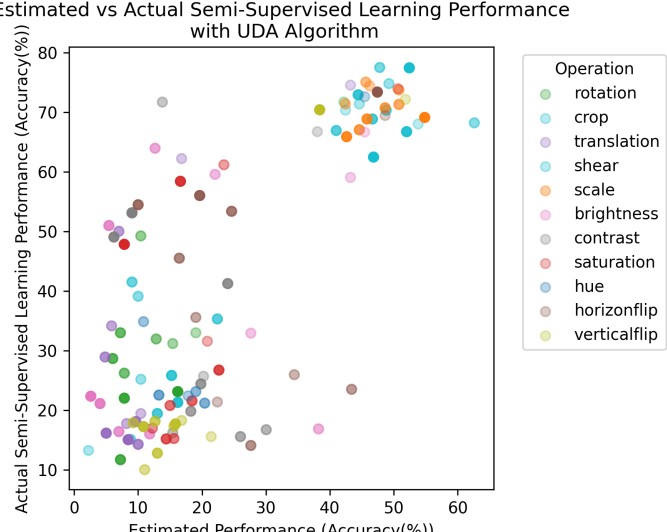

*Figure 8.* This figure illustrates the comparison between estimated performance and final actual performance after full semi-supervised learning on CIFAR-10 with the UDA Algorithm.

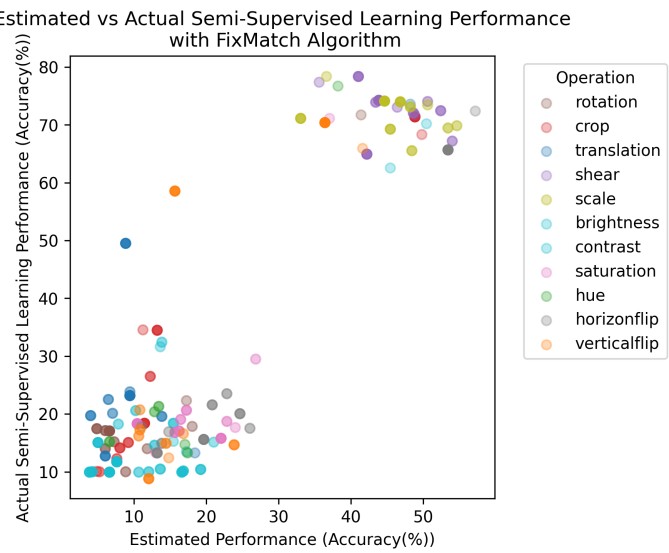

*Figure 9.* This figure illustrates the comparison between estimated performance and final actual performance after full semi-supervised learning on CIFAR-10 with the FixMatch Algorithm.

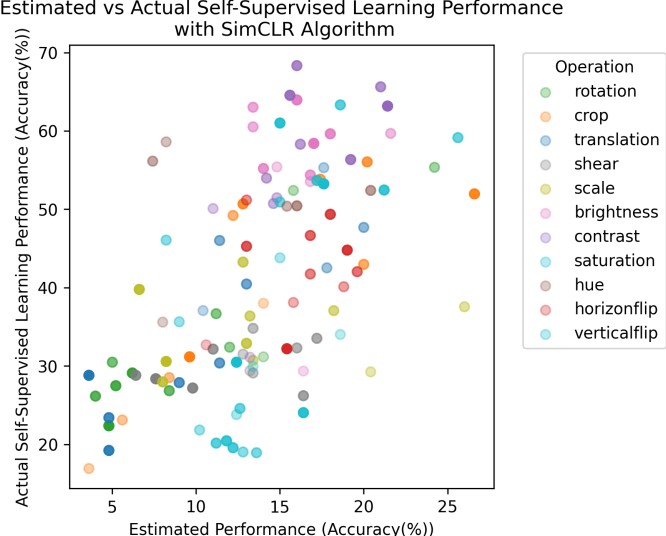

*Figure 10.* This figure illustrates the comparison between estimated performance and final actual performance after full self-supervised learning on CIFAR-10 with the SimCLR Algorithm.

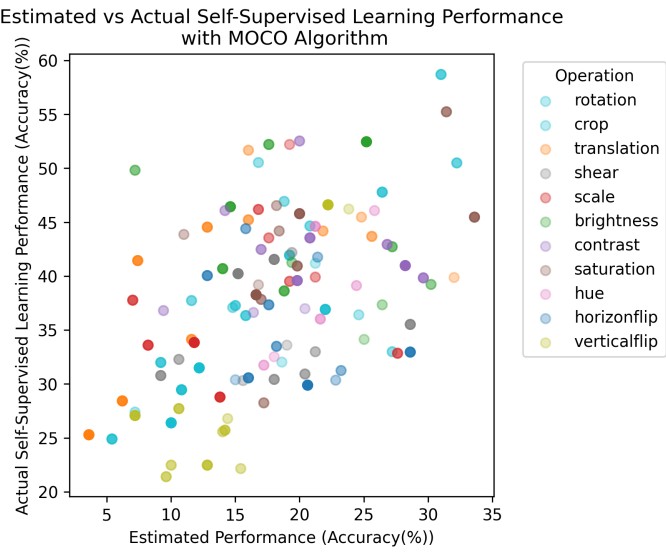

*Figure 11.* This figure illustrates the comparison between estimated performance and final actual performance after full self-supervised learning on CIFAR-10 with the MOCO Algorithm.

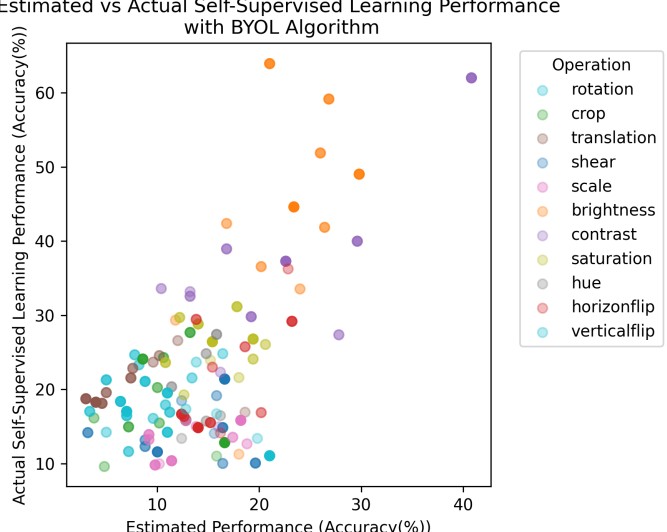

*Figure 12.* This figure illustrates the comparison between estimated performance and final actual performance after full self-supervised learning on CIFAR-10 with the BYOL Algorithm.

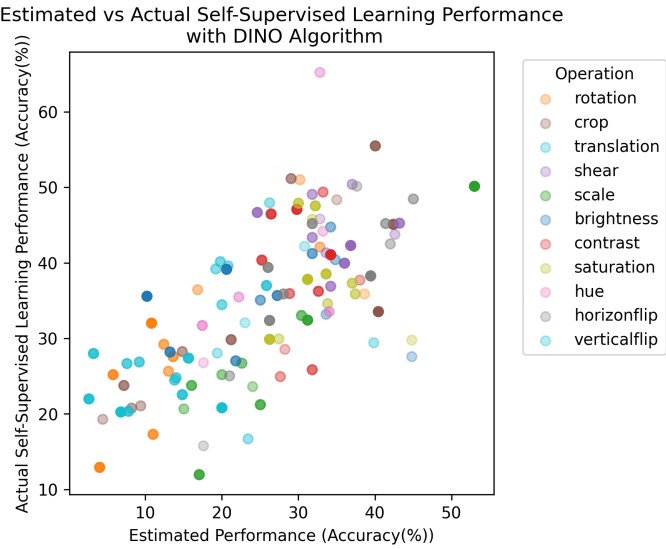

*Figure 13.* This figure illustrates the comparison between estimated performance and final actual performance after full self-supervised learning on CIFAR-10 with the DINO Algorithm.

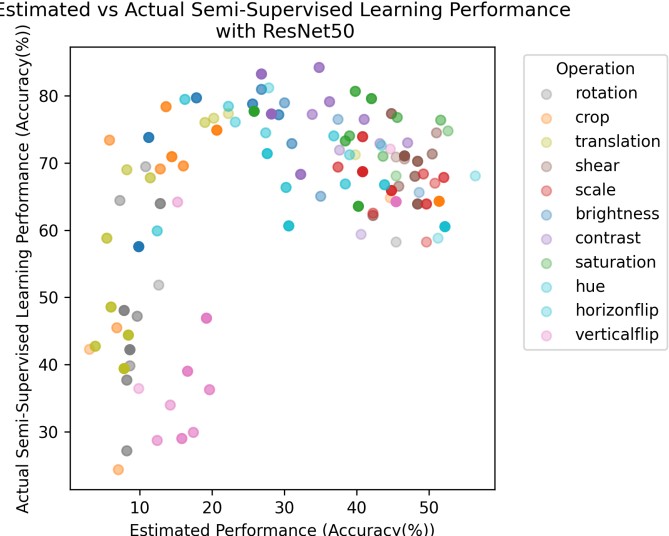

*Figure 14.* This figure illustrates the comparison between estimated performance and final actual performance after full semi-supervised learning on CIFAR-10 with the ResNet-50 backbone.

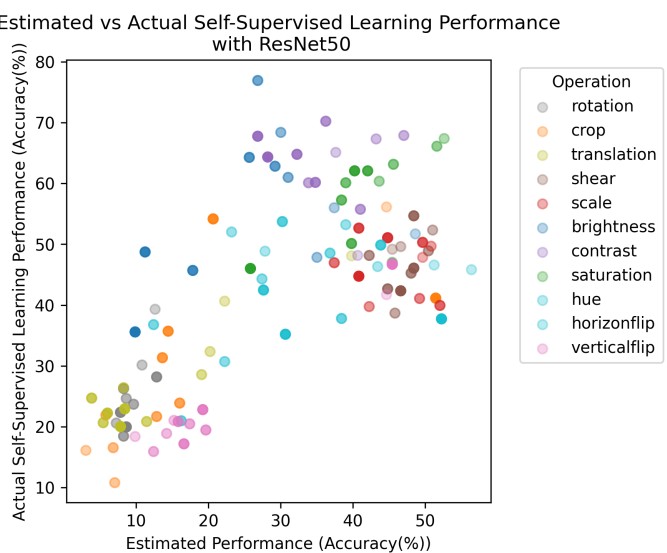

*Figure 15.* This figure illustrates the comparison between estimated performance and final actual performance after full self-supervised learning on CIFAR-10 with the ResNet-50 backbone.

performance and actual performance under the self-supervised setting, which is 0.674. The comparison results can be seen in Figure 16.

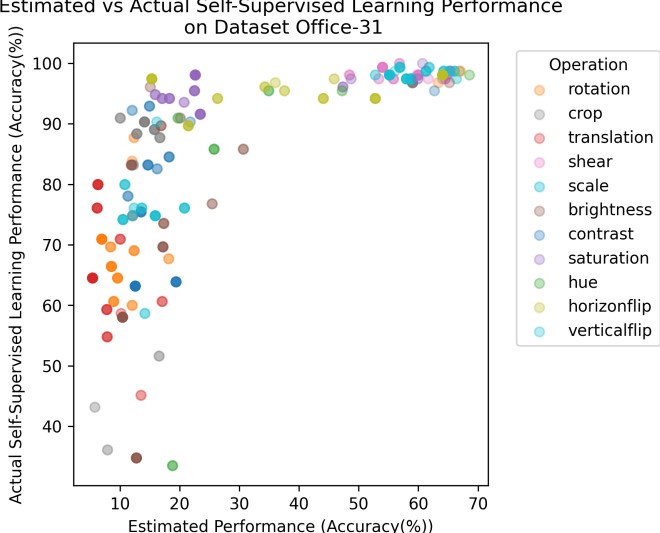

*Figure 16.* This figure illustrates the comparison between estimated performance and final actual performance after full self-supervised learning on Office-31.

## F. Experiments on the Data Scale Robustness

In the basic experiments, considering the estimation efficiency, we only used unlabeled data that was 10 times the size of the labeled data for performance estimation. To demonstrate that using more unlabeled data can achieve more accurate performance estimation, we conducted additional experiments under the setting where the unlabeled data is 50 times the size of the labeled data. We also plotted scatter plots and calculated the correlation coefficients between estimated performance and actual performance under semi-supervised and self-supervised settings, which are 0.803 and 0.854 respectively. The comparison results can be seen in Figures 17 and 18.

## G. Experiments on the Hyperparameters Robustness

To verify that the effectiveness of our proposed estimation scheme is not excessively affected by training hyperparameters, we conducted further validation on two key hyperparameters: the number of epochs and the learning rate.

In the basic settings, the number of epochs was set to 5; we further validated the effectiveness of the method when the number of epochs was 10. We also plotted scatter plots and calculated the correlation coefficients between estimated performance and actual performance under semi-supervised and self-supervised settings, which are 0.874 and 0.891 respectively. The comparison results can be seen in Figures 19 and 20.

In the basic settings, the learning rate was set to 5e-5; we further validated the effectiveness of the method when the learning rate was 1e-5. We also plotted scatter plots and calculated the correlation coefficients between estimated performance and actual performance under semi-supervised and self-supervised settings, which are 0.666 and 0.944 respectively. The comparison results can be seen in Figures 21 and 22.

## H. Cost Comparison

We compared the resources consumed by our proposed estimation method and the post-training validation method in evaluating the impact of unsupervised pretext tasks. Specifically, we compared the computation time, the amount of labeled data, and the amount of unlabeled data required to assess whether a pretext task is suitable for the target task, demonstrating

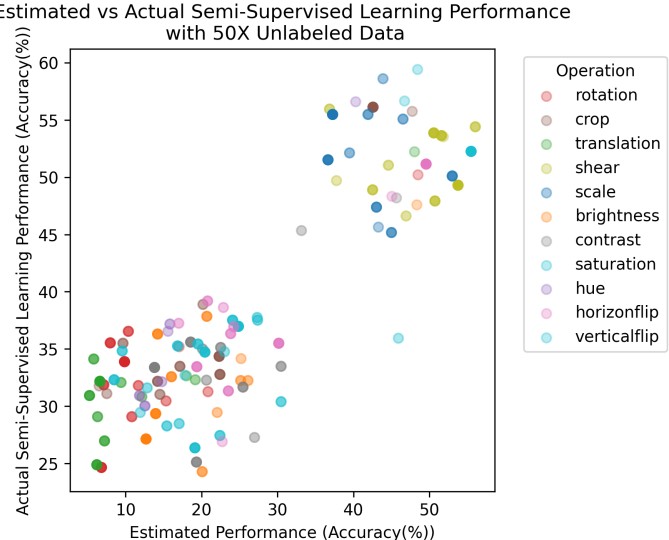

*Figure 17.* This figure illustrates the comparison between estimated performance and final actual performance after full semi-supervised learning on CIFAR-10 with 50× unlabeled data.

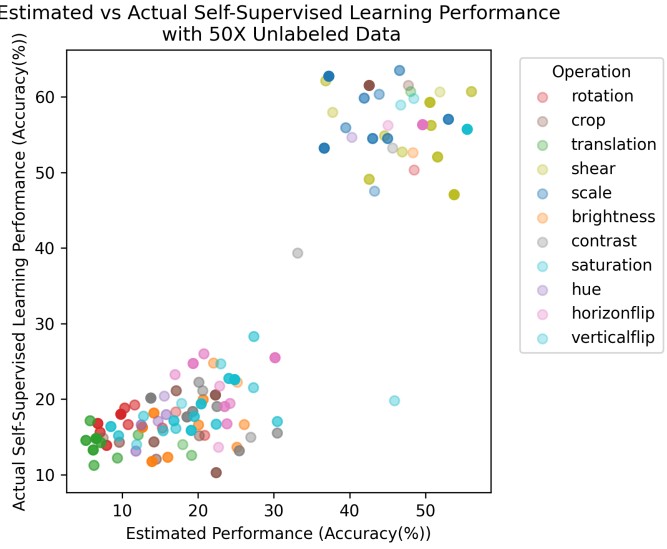

*Figure 18.* This figure illustrates the comparison between estimated performance and final actual performance after full self-supervised learning on CIFAR-10 with 50× unlabeled data.

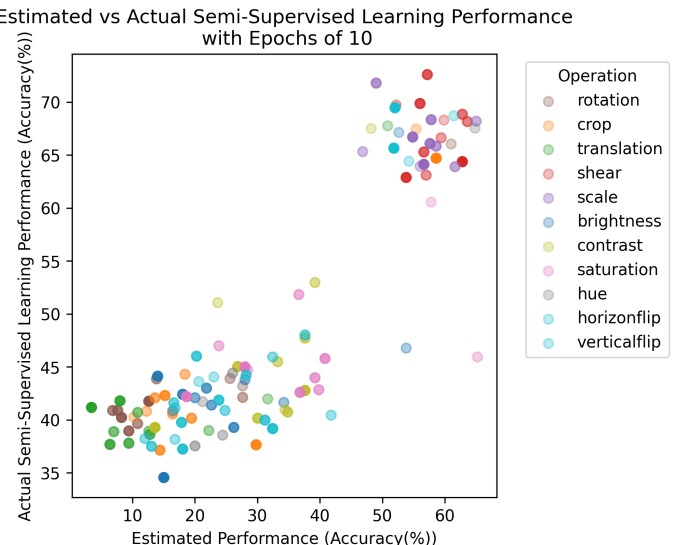

*Figure 19.* This figure illustrates the comparison between estimated performance and final actual performance after full semi-supervised learning on CIFAR-10 with the epoch of 10.

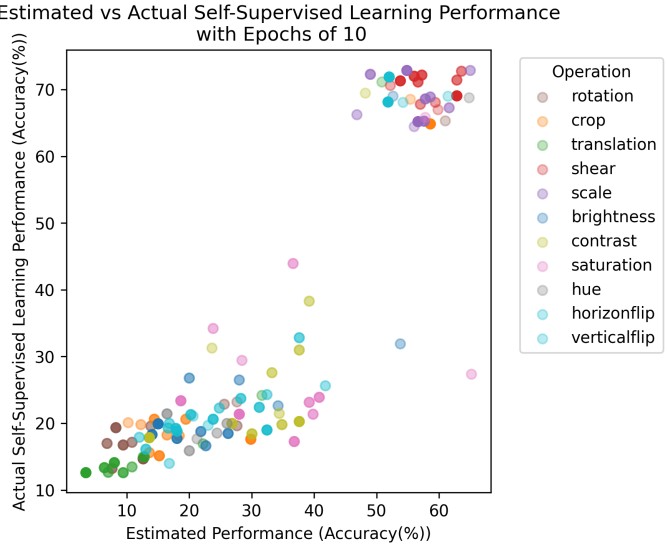

*Figure 20.* This figure illustrates the comparison between estimated performance and final actual performance after full self-supervised learning on CIFAR-10 with the epoch of 10.

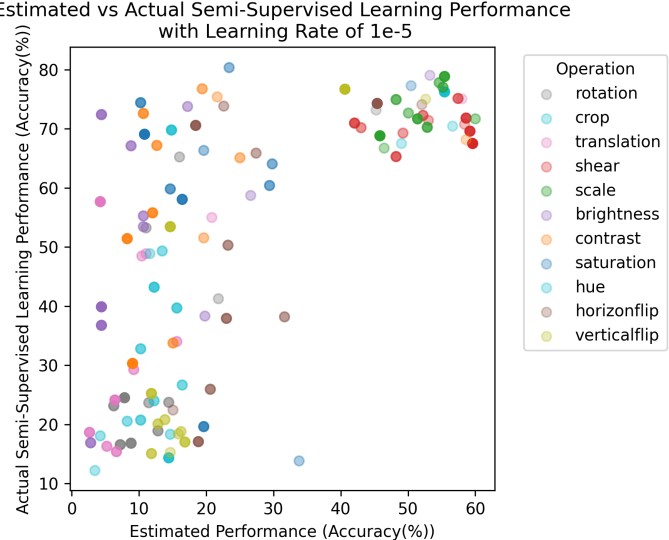

*Figure 21.* This figure illustrates the comparison between estimated performance and final actual performance after full semi-supervised learning on CIFAR-10 with the learning rate of 1e-5.

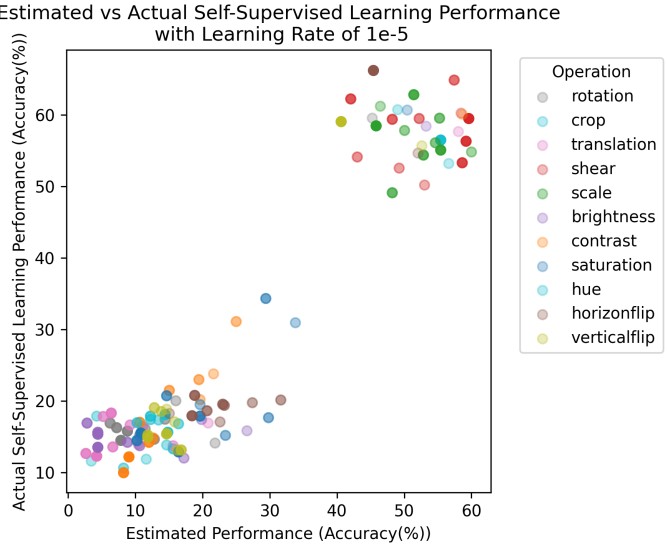

*Figure 22.* This figure illustrates the comparison between estimated performance and final actual performance after full self-supervised learning on CIFAR-10 with the learning rate of 1e-5.

the significant superiority of our proposed estimation method in terms of resource consumption. The time is uniformly reported as the number of seconds run on a single A800 GPU. The results can be seen in Table 4.

*Table 4.* Comparison of resource consumption between the actual performance obtained through training&validation and the estimated performance derived from our proposed method on the CIFAR-10 dataset.

| Method | Time | labeled data | unlabeled data |
|---|---|---|---|
| Semi-Supervised Training and Validation | 1547.3 | 50(training)+10000(validation) | 49950(training) |
| Self-Supervised Training and Validation | 1651.6 | 50(training)+10000(validation) | 49950(training) |
| Estimation | 29.6 | 50(estimation) | 500(estimation) |

# I. Proofs of Estimation Error Bounds

In this subsection, we provide the proofs of the estimation error bounds presented above. The analysis mainly relies on the uniform convergence property induced by Rademacher complexity and standard concentration inequalities.

*Proof of Theorem 1.* Consider the learnability estimation risk $R_{\text{unlearnable}}$ and its empirical estimator $\hat{R}_{\text{unlearnable}}$. Assume that the associated loss function $\ell_{\text{pretext}}(\cdot)$ is bounded in $[0, 1]$.

According to the standard Rademacher complexity generalization bound, for any hypothesis space $\mathcal{F}_{\text{pretext}}$ and any $\delta \in (0, 1)$, with probability at least $1 - \delta$, the following uniform convergence inequality holds:

$$\sup_{f \in \mathcal{F}_{\text{pretext}}} \left| \mathbb{E}[\ell_{\text{pretext}}(f)] - \frac{1}{n_U} \sum_{i=1}^{n_U} \ell_{\text{pretext}}(f(x_i)) \right| \leq 2\Re(\mathcal{F}_{\text{pretext}}, n_U) + \sqrt{\frac{\log(2/\delta)}{2n_U}}. \tag{24}$$

Substituting $f = \hat{f}_{\text{pretext}}$ into the above inequality directly yields

$$|\hat{R}_{\text{unlearnable}} - R_{\text{unlearnable}}| \leq 2\Re(\mathcal{F}_{\text{pretext}}, n_U) + \sqrt{\frac{\log(2/\delta)}{2n_U}}. \tag{25}$$

Defining

$$\epsilon_{\text{pretext}} = 2\Re(\mathcal{F}_{\text{pretext}}, n_U) + \sqrt{\frac{\log(2/\delta)}{2n_U}}, \tag{26}$$

we complete the proof. □

*Proof of Theorem 2.* The proof follows the same argument as Theorem 1. Assume that the target loss $\ell_{\text{target}}(\cdot)$ is bounded in $[0, 1]$.

Applying the Rademacher complexity bound to the target hypothesis space $\mathcal{F}_{\text{target}}$, for any $\delta \in (0, 1)$, with probability at least $1 - \delta$, we have

$$\sup_{f \in \mathcal{F}_{\text{target}}} \left| \mathbb{E}[\ell_{\text{target}}(f)] - \frac{1}{n_L} \sum_{i=1}^{n_L} \ell_{\text{target}}(f(x_i, y_i)) \right| \leq 2\Re(\mathcal{F}_{\text{target}}, n_L) + \sqrt{\frac{\log(2/\delta)}{2n_L}}. \tag{27}$$

Substituting $f = \hat{f}_{\text{target}}$ gives

$$|\hat{R}_{\text{unreliable}} - R_{\text{unreliable}}| \leq 2\Re(\mathcal{F}_{\text{target}}, n_L) + \sqrt{\frac{\log(2/\delta)}{2n_L}}. \tag{28}$$

Defining

$$\epsilon_{\text{target}} = 2\mathfrak{R}(\mathcal{F}_{\text{target}}, n_L) + \sqrt{\frac{\log(2/\delta)}{2n_L}}, \tag{29}$$

the theorem is proved. $\square$

*Proof of Theorem 3.* The incompleteness metric jointly depends on the pretext estimator and the target estimator. Using the triangle inequality, we obtain

$$|\hat{R}_{\text{incomplete}} - R_{\text{incomplete}}| \leq |\hat{R}_{\text{pretext}} - R_{\text{pretext}}| + |\hat{R}_{\text{target}} - R_{\text{target}}|. \tag{30}$$

Applying the bounds established in Theorem 1 and Theorem 2 yields

$$|\hat{R}_{\text{incomplete}} - R_{\text{incomplete}}| \leq \epsilon_{\text{pretext}} + \epsilon_{\text{target}}. \tag{31}$$

Thus, the proof is completed. $\square$

*Proof of Theorem 4.* Assume that the target risk can be expressed as a smooth function

$$R_{\text{target}} = g(R_{\text{unlearnable}}, R_{\text{unreliable}}, R_{\text{incomplete}}), \tag{32}$$

and its empirical counterpart is

$$\hat{R}_{\text{target}} = g(\hat{R}_{\text{unlearnable}}, \hat{R}_{\text{unreliable}}, \hat{R}_{\text{incomplete}}). \tag{33}$$

Define the perturbations

$$\Delta_u = \hat{R}_{\text{unlearnable}} - R_{\text{unlearnable}}, \tag{34}$$

$$\Delta_r = \hat{R}_{\text{unreliable}} - R_{\text{unreliable}}, \tag{35}$$

$$\Delta_c = \hat{R}_{\text{incomplete}} - R_{\text{incomplete}}. \tag{36}$$

According to Theorem 1–3, with probability at least $1 - \delta$, we have

$$|\Delta_u| \leq \epsilon_{\text{pretext}}, \tag{37}$$

$$|\Delta_r| \leq \epsilon_{\text{target}}, \tag{38}$$

$$|\Delta_c| \leq \epsilon_{\text{pretext}} + \epsilon_{\text{target}}. \tag{39}$$

Applying the third-order Taylor expansion of $g(\cdot)$ around $(R_{\text{unlearnable}}, R_{\text{unreliable}}, R_{\text{incomplete}})$, we obtain

$$\hat{R}_{\text{target}} - R_{\text{target}} = \sum_i \frac{\partial g}{\partial R_i} \Delta_i + \frac{1}{2} \sum_{i,j} \frac{\partial^2 g}{\partial R_i \partial R_j} \Delta_i \Delta_j$$
$$+ \frac{1}{6} \sum_{i,j,k} \frac{\partial^3 g}{\partial R_i \partial R_j \partial R_k} \Delta_i \Delta_j \Delta_k + \mathcal{O}(\|\Delta\|^4), \tag{40}$$

where $i, j, k \in \{u, r, c\}$.

Assuming that the derivatives of $g$ are bounded by constants absorbed into the inequality coefficients, the first-order term is bounded by

$$\mathcal{O}(\|\Delta\|) \leq 2(\epsilon_{\text{pretext}} + \epsilon_{\text{target}}). \tag{41}$$

Similarly, the second-order term satisfies

$$\mathcal{O}(\|\Delta\|^2) \leq (\epsilon_{\text{pretext}} + \epsilon_{\text{target}})^2 + \epsilon_{\text{pretext}} \cdot \epsilon_{\text{target}}, \tag{42}$$

and the dominant third-order interaction term satisfies

$$\mathcal{O}(\|\Delta\|^3) \leq \epsilon_{\text{pretext}} \cdot \epsilon_{\text{target}} \cdot (\epsilon_{\text{pretext}} + \epsilon_{\text{target}}). \tag{43}$$

Combining the first-, second-, and third-order terms yields

$$\begin{aligned}
|\hat{R}_{\text{target}} - R_{\text{target}}| \leq\ & 2(\epsilon_{\text{pretext}} + \epsilon_{\text{target}}) \\
& + (\epsilon_{\text{pretext}} + \epsilon_{\text{target}})^2 \\
& + \epsilon_{\text{pretext}} \cdot \epsilon_{\text{target}} \cdot (\epsilon_{\text{pretext}} + \epsilon_{\text{target}}).
\end{aligned} \tag{44}$$

Therefore, the theorem is proved. $\square$

## J. Future Work

It is not difficult to foresee that in the future, the scale of models, data and targets will continue to grow, while reliable labels will not increase accordingly. This means that unsupervised training will occupy an increasingly significant role. As the starting point of training, the selection of pretext tasks based on prior knowledge or assumption will become increasingly important throughout the machine learning pipeline. In the future, we plan to increase our efforts in the design of unsupervised tasks, aiming to enable models to learn more complex prior assumption from unlabeled data, especially under more complex models, data, and targets. More importantly, we propose empirical methods to quantify the theoretical indicators in applications. These methods enable low-cost estimation of model's potential performance on the target task before training and validation begins. This makes it feasible to rapidly assess the benefit of unsupervised pretext tasks in practical scenarios, dramatically reducing trial-and-error risks and costs, and shifting task selection from heuristic-based to theory-based.

